# *Amaranthus cruentus* L. Seed Oil Counteracts UVA-Radiation-Induced Inhibition of Collagen Biosynthesis and Wound Healing in Human Skin Fibroblasts

**DOI:** 10.3390/ijms25020925

**Published:** 2024-01-11

**Authors:** Katarzyna Wolosik, Magda Chalecka, Jerzy Palka, Blanka Mitera, Arkadiusz Surazynski

**Affiliations:** 1Department of Cosmetology, Medical University of Bialystok, Kilinskiego 1, 15-089 Bialystok, Poland; katarzyna.wolosik@umb.edu.pl; 2Department of Medicinal Chemistry, Medical University of Bialystok, Kilinskiego 1, 15-089 Bialystok, Poland; magda.chalecka@umb.edu.pl (M.C.); pal@umb.edu.pl (J.P.); blanka.mitera@gmail.com (B.M.)

**Keywords:** UVA radiation, dermal fibroblasts, collagen biosynthesis, wound healing, *Amaranthus cruentus* L. seed oil, sun-protective substance

## Abstract

The effect of *Amaranthus cruentus* L. seed oil (AmO) on collagen biosynthesis and wound healing was studied in cultured human dermal fibroblasts exposed to UVA radiation. It was found that UVA radiation inhibited collagen biosynthesis, prolidase activity, and expression of the β1-integrin receptor, and phosphorylated ERK1/2 and TGF-β, while increasing the expression of p38 kinase. The AmO at 0.05–0.15% counteracted the above effects induced by UVA radiation in fibroblasts. UVA radiation also induced the expression and nuclear translocation of the pro-inflammatory NF-κB factor and enhanced the COX-2 expression. AmO effectively suppressed the expression of these pro-inflammatory factors induced by UVA radiation. Expressions of β1 integrin and IGF-I receptors were decreased in the fibroblasts exposed to UVA radiation, while AmO counteracted the effects. Furthermore, AmO stimulated the fibroblast’s migration in a wound healing model, thus facilitating the repair process following exposure of fibroblasts to UVA radiation. These data suggest the potential of AmO to counteract UVA-induced skin damage.

## 1. Introduction

Solar radiation is a significant factor contributing to the acceleration of skin aging through impairment of metabolism in the skin cells. Two mechanisms of these harmful effects are recognized, and the first involves the production of reactive oxygen species (ROS), the second, UV-radiation-induced DNA damage. Both processes contribute to deregulation of metabolism in the skin cells, inhibition of collagen biosynthesis, and induction of cell death through apoptosis [1,2]. As a result, collagen metabolism is impaired at the transcriptional and post-transcriptional level. At the transcriptional level, UVA induces NF-κB expression, the inhibitor of collagen gene transcription [3]. Post-transcriptionally, it impairs prolidase activity, the enzyme providing proline from imidodipeptides for collagen biosynthesis [4], and activates matrix metalloproteinases (MMPs) that degrade collagen extracellularly [5]. In addition, collagen also plays a critical role in regulating cellular metabolism as a ligand of integrin receptors. Integrin receptors participate in signaling that regulates collagen biosynthesis and prolidase activity [4]. Both processes are stimulated by insulin-like growth factor-I (IGF-I), the most potent factor stimulating collagen biosynthesis [6]. Under physiological conditions, the activation of β1-integrin and IGF-I receptors initiates a cascade of the mitogen-activated protein kinase (MAP-kinase) signaling pathway, including extracellular signal-regulated kinases (ERK-1 and ERK-2) [4]. It is well established that ROS down-regulate expression of MAP kinase ERK1/2 and support expression of stress-activated kinases, JNK (c-Jun N-terminal kinase) and p38. The effectors of MAP kinases are transcription factors, including c-Jun and c-Fos, which form the activating protein complex AP-1 (activating protein-1). This protein plays an important role in regulation of collagen metabolism as an inhibitor of pro-collagen type I gene expression and the transforming growth factor beta (TGF-β) signaling pathway [7,8,9]. Chronic exposure to UVA induces inflammation in the skin, resulting in the release of cytokines and proinflammatory factors, including the nuclear transcription factor NF-κB (nuclear factor kappa B) [5,10,11]. Activated NF-κB plays an important role in regulating the expression of various genes involved in immune and inflammatory responses, including the expression of cyclooxygenase COX-2, an enzyme associated with the inflammatory process [12].

Seed oil derived from *Amaranthus cruentus* L. (AmO) is considered a promising source of bioactive compounds counteracting UVA-induced skin damage [13]. The grains of *Amaranthus cruentus* L. cultivated for their seeds exhibit a greater lipid content (7.2%) compared to traditional cereal plants. The lipid content is more than that found in wheat (1.9%), rye (1.9%), or rice (2.1%) grains, and maize (4.5%). The seeds contain crude fat ranging from 5.6% to as much as 10.9%. Not only is the quantity favorable, but also the composition of lipids in amaranth grain is advantageous [14]. The oil derived from *Amaranthus cruentus* L. was analyzed by León-Camacho et al. [15] to identify both major and minor compounds. Fatty acids, triglycerides, sterols, methyl sterols, terpenic and aliphatic alcohols, tocopherols, and hydrocarbons have been meticulously identified using standards and mass spectrometry. The content of these compounds, along with the equivalent carbon numbers (ECNs) and triglyceride carbon numbers (TCNs), has been compared with the findings of other edible vegetable and cereal oils such as canola, coconut, cottonseed maize, palm, soya, sunflower, and others. The hydrocarbon composition is particularly noteworthy, with a predominant presence of squalene (4.16 g/kg of seed) and a concentration of n-alkenes (C23:1–C33:1) reaching 332 ppm (parts per million) [15].

The oil composition is predominantly characterized by unsaturated fatty acids, specifically oleic acid, linoleic acid, and linolenic acid, which collectively contribute to diverse health-promoting effects. Certain unsaturated fatty acids, notably linolenic acid, are exogenous acids indispensable for the human body, serving a crucial role in hormone synthesis, cellular membrane structure, and the regulation of membrane permeability [14]. The findings in Martirosyan et al.’s [16] study indicate that Amaranthus oil has the potential to lower cholesterol levels in the bloodstream, making it a viable candidate for inclusion in functional food products designed for the prevention and treatment of cardiovascular diseases. Adopting a diet enriched with Amaranthus oil may contribute to lowering blood pressure, presenting itself as a promising alternative to pharmacological interventions for individuals with hypertension. Notably, the study suggests that the synergistic combination of Amaranthus oil with a hypo-sodium antiatherogenic diet is more effective in reducing blood cholesterol levels compared to solely the hypo-sodium antiatherogenic diet. These insights emphasize the potential of incorporating Amaranthus oil into dietary strategies for cardiovascular health [16].

Due to its abundant nutritional properties, certain Amaranthus preparations have found applications in the cosmetics industry for hair and skin health, providing exceptional moisturizing capabilities, soothing irritations, promoting accelerated wound healing, and demonstrating antimicrobial properties. Notably, Amaranthus seed oil contributes to skin nourishment and exhibits anti-aging effects, facilitating the regeneration, nourishment, and strengthening of the epidermis. Functioning as an antioxidant, it offers protective qualities [17]. Lacatusu and colleagues explored innovative sunscreen formulations incorporating nanostructured lipid carriers (NLCs) as delivery systems for an antioxidant and anti-UV bioactive. In this study, Amaranthus oil, along with pumpkin seed oil, was integrated into the lipid NLCs’ core, creating novel delivery systems capable of simultaneously encapsulating UVA and UVB filters and an antioxidant. This pioneering approach represents a non-invasive design for herbal cosmetic formulations, delivering superior photoprotection and enhanced antioxidant properties [17,18].

*Amaranthus cruentus* L. seed oil contains substantial amounts of squalene. Depending on the variety, the oil extracted from these seeds can contain up to 8% of squalene. As a natural component of human skin lipids, squalene plays a crucial role in skin physiology, constituting approximately 13% of human sebum [19]. Squalene is characterized by high biological activity; it serves as a potent antioxidant, and a natural emollient, and also reduces water loss from the epidermis, thereby influencing skin hydration. This property is utilized in preparations for treating skin conditions such as atopic dermatitis or psoriasis. Squalene also restricts the development of acne lesions in acne vulgaris [19]. In contemporary biomedical and cosmetic applications, squalene and its hydrogenated derivative, squalane, are employed in pharmaceutical and dermatological formulations for addressing skin disorders, leveraging their emollient and moisturizing properties. Furthermore, squalene has been incorporated into antiviral emulsions and utilized as an adjuvant in vaccines, and is traditionally employed in China as agents with anti-fatigue and anti-aging attributes [15].

The daily intake of squalene among individuals in Mediterranean countries can vary from 30 mg to 200 mg and even up to 1000 mg. The recommended minimum daily intake is 11 mg [19,20]. Therapeutic doses of squalene, ranging from 2000 to 5000 mg per day, are considered effective against cancer. This leads to a reduced risk of bowel, breast, and skin cancer [19,21]. Squalene demonstrates the ability to inhibit tumor cell growth and induce regression in existing tumors [17,22,23,24]. In recent years, the protective role of squalene has been demonstrated against various carcinogens, including those associated with conditions like cell leukemia [24] or skin cancer [25]. Squalene also enhances the effects of antineoplastic agents such as adriamycin, 5-fluorouracil, or bleomycin [26]. In addition to alkylglycerols and omega-3s, squalene exerts a beneficial effect on the natural immune system and can be helpful in the treatment of diseases associated with impaired immune function [19,26,27].

We found that AmO at 0.05%, 0.1%, and 0.15% concentrations evoked a protective effect on decreased viability of fibroblast exposure to UVA irradiation [13]. Exposure to UVA radiation at a dosage of 10 J/cm^2^ resulted in the down-regulation of p-AKT and mTOR expression, alongside the upregulation of p53, caspase-3, caspase-9, and PARP expression. This led to a substantial reduction in the viability of human skin fibroblasts, triggering apoptosis. Treatment of UVA-exposed fibroblasts with *Amaranthus cruentus* seed oil (AmO) at concentrations of 0.1% and 0.15% mitigated the expression of apoptosis-related markers (p53, caspase-3, caspase-9, and PARP) and reinstated the expression of p-Akt and mTOR proteins. The underlying molecular mechanism of this phenomenon is linked to the stimulation of antioxidant processes through the activation of Nrf2. This implies that AmO stimulates the antioxidant system in fibroblast cells, counteracting the effects of UVA-induced oxidative stress [13]. These data led us to evaluate the effect of AmO on collagen metabolism in fibroblasts exposed to UVA radiation.

The effect of AmO on collagen metabolism and wound healing in cells exposed to UVA radiation is currently unknown. Given the oil’s rich composition of bioactive compounds, it has the potential to exhibit protective effects not only against oxidative stress but also other metabolic disturbances induced by UV radiation. This study aims to investigate the effect of AmO on the collagen biosynthesis and wound healing process in human skin fibroblasts exposed to UVA radiation.

## 2. Results and Discussion

Prolonged exposure of skin to UV radiation leads to generation of reactive oxygen species (ROS) in the skin cells. They cause a redox imbalance, a deregulation of collagen synthesis, and damage to DNA, proteins, and lipids. They also affect the integrity of biological membranes of skin cells, fibroblasts, and keratinocytes [5]. In our previous study, we found that exposure of fibroblasts to different doses of UVA radiation contributed to a decrease in the cell viability in a UVA-dose-dependent manner [13]. Based on these data, in the present study, we selected a dosage of 10 J/cm^2^, which was the IC50 value for cell viability in our experimental protocols [13], to evaluate the effect of *Amaranthus cruentus* L. seed oil (AmO) on collagen metabolism and wound healing.

Plant oils are among the cosmetic materials that demonstrate beneficial effects on the skin, due to their chemical composition, primarily the presence of Essential Polyunsaturated Fatty Acids (PUFAs), as well as other components of the unsaponifiable fraction, such as squalene, phytosterols, and tocopherols [28]. AmO, in this regard, is characterized by a favorable chemical composition, containing approximately 60% linoleic acid, 8–10% squalene, and tocopherols, thus demonstrating protective and antioxidant potential against UV radiation. In our previous study, we demonstrated that fibroblast viability decreased to approximately 55% of the control value when exposed to a 10 J/cm^2^ UVA radiation dose.

Collagen is the main structural protein of connective tissue, providing the skin with proper tension, elasticity, and flexibility. Excessive exposure to the sun, extreme temperatures, and certain compounds in cosmetics can affect the structure of collagen fibers, resulting in a worsened skin condition [29]. Collagen biosynthesis was evaluated using measurement of L-5-[^3^H]-proline incorporation into collagen proteins. The results were expressed as dpm of L-5-[^3^H]-proline released from collagenase-sensitive proteins per milligram of total protein in the homogenate extract, and expressed as a percentage of the control value (100%) (Figure 1).

We found that AmO at studied concentrations slightly inhibited collagen biosynthesis in fibroblasts; however, the inhibition was statistically insignificant. When the cells were exposed to UVA radiation, collagen biosynthesis was decreased to 58% of the control value (statistically significant at *p* < 0.05). AmO at concentrations of 0.05%, 0.1%, and 0.15% partially counteracted UVA-dependent inhibition of collagen biosynthesis to about 78%, 81%, and 85% of the control value, respectively. The findings indicate that the application of AmO after UVA exposure led to more favorable outcomes compared to UVA alone. UVA alone resulted in a reduction in collagen biosynthesis to 58% of the control value. However, the differences observed were about 20%, 23%, and 27% higher for AmO concentrations of 0.05%, 0.1%, and 0.15%, respectively, than in UVA-only-treated cells. These differences document a significant improvement in the effectiveness of collagen biosynthesis after the application of AmO following UVA exposure compared to UVA exposure alone. These data suggest that AmO has a protective effect on collagen biosynthesis in human skin fibroblasts exposed to UVA radiation.

Collagen content in the tissues depends on the balance between collagen biosynthesis and degradation. An important regulator of collagen turnover is prolidase, a cytoplasmic enzyme catalyzing the final stage of collagen degradation. It cleaves di- and tri-peptides containing C-terminal proline. Released proline is reused for collagen biosynthesis and cell growth [30]. Therefore, we decided to study the prolidase activity in human skin fibroblasts exposed to UVA radiation and treated with AmO. We assessed this activity using a colorimetric method, based on the measurement of proline released from the substrate (Gly-Pro), using Chinard’s reagent. In UVA-treated cells, prolidase activity was decreased to 54% of the control value. AmO at concentrations of 0.05%, 0.1%, and 0.15% partially counteracted UVA-dependent inhibition of prolidase activity to about 75%, 80%, and 85% of the control value, respectively. The findings indicate that the application of AmO after UVA exposure led to more favorable outcomes compared to UVA alone. UVA alone resulted in a reduction in prolidase activity to 54% of the control value. However, the differences observed were about 21%, 26%, and 31% higher for concentrations of 0.05%, 0.1%, and 0.15%, respectively, than in UVA-only-treated cells. These data suggest a significant increase in prolidase activity after the application of AmO following UVA exposure, compared to UVA exposure alone (Figure 2). These data show the protective effect of the AmO on UVA-induced impairment of collagen biosynthesis and prolidase activity in fibroblasts.

Collagen plays a significant role in cellular metabolism as a ligand of integrin receptors containing a β1 subunit. The β1-integrin receptor is involved in signaling pathways that regulate collagen biosynthesis and prolidase activity. Interactions between collagen and these receptors activate intracellular signaling pathways, contributing to the regulation of various cellular metabolic functions [4].

AmO at 0.05%, 0.1%, and 0.15% concentrations slightly decreased (to about 85% of control) the expression of the β1-integrin receptor as presented in Figure 3A. In UVA-treated cells, the expression of this receptor was drastically decreased, while the application of AmO at a concentration of 0.1% and 0.15% counteracted this process and it was statistically significant. Regarding the results of collagen biosynthesis and prolidase activity, there is another level of correlation between these processes and β1-integrin receptor expression in UVA- and AmO-treated cells. It suggests that UVA radiation, by decreasing the expression of the β1-integrin receptor, down-regulates prolidase activity, leading to reduced collagen production. The protective effect of AmO on the UVA-induced down-regulation of β1-integrin expression may represent a molecular mechanism for protection of collagen biosynthesis against the deleterious effect of UV radiation on this process.

Prolidase activity and collagen biosynthesis are also regulated by insulin-like growth factor I receptor (IGF-IR) signaling. IGF-I belongs to peptide growth factors that play a significant role in cell metabolism, proliferation, and differentiation, and it is the most potent stimulator of collagen biosynthesis [6]. The expression of IGF-I is elevated in skin cells under inflammatory conditions. For instance, in psoriatic skin, fibroblasts evoke a high expression of IGF-I [31].

To date, the effect of UVA on IGF-IR expression in cells of normal skin is not well recognized. The study presented in this report shows that fibroblasts exposed to UVA radiation evoke decreased IGF-IR receptor expression to about 25% of the control, suggesting the mechanism for UVA-induced inhibition of collagen biosynthesis. The application of AmO at tested concentrations counteracted the deleterious effect of UVA on IGF-IR expression to 57%, 64%, and 66% of the control (Figure 3B). These findings suggest that down-regulation of IGF-IR signaling may play an important role in the mechanism of UVA-induced inhibition of collagen biosynthesis. They also suggest the mechanism for protective action of AmO on this process.

Furthermore, a drastic increase in the expression of these receptors is observed following AmO treatment in the UVA-exposed cells, compared to the cells subjected to UVA alone. Specifically, at 0.1% and 0.15% of AmO, the expression of the β1-integrin receptor was increased to about 50% of the control and the increase was statistically significant. Similarly, at 0.05%, 0.1%, and 0.15% of AmO, the expression of the IGF-IR receptor was increased to 57%, 64%, and 66% of the control, respectively.

The observed upregulation of expressions of the β1-integrin receptor and IGF-IR following AmO treatment in UVA-exposed cells suggests a potential regulatory mechanism of AmO on cellular signaling. These findings provide rationale for further investigations on the AmO-induced modulation of the cell receptor dynamics.

We confirmed the potential involvement of the β1-integrin and IGF-I receptors in the molecular mechanism through which UVA affects collagen biosynthesis and prolidase activity. It is well established that stimulation of these receptors leads to the activation of mitogen-activated protein kinases (MAPKs) and transforming growth factor β (TGF-β) [8,9]. MAPK signaling pathways play an important role in cellular metabolism. For instance, in skin cells, ROS induce a decrease in the expression of phosphorylated forms of ERK 1/2 kinase (p-ERK 1/2) and an increase in the expression of stress-activated kinase p38, leading to the upregulation of expression of transcription factors, including c-Jun and c-Fos, which form the activator protein-1 (AP-1) complex [7,8,9]. This protein is a key regulator in various skin processes, including photoaging, by affecting the TGF-β signaling pathway and inhibiting the expression of the type I procollagen gene. In our previous studies, we demonstrated that UVA radiation induced an increase in ROS generation in fibroblasts [13]. In present studies, we show that UVA induced a decrease in the expression of p-ERK 1/2 (Figure 4) and TGF-β (Figure 5A) and an increase in p38 protein expression (Figure 5B). This confirms the hypothesis on the correlation between UVA-induced ROS generation, deregulation of MAPK-dependent signaling pathways, and collagen biosynthesis in fibroblasts. From several other studies, it is evident that the application of antioxidants from natural sources (e.g., green tea, pomegranate extract) reverses the negative effects (deregulation of the above-described cellular signaling pathways) of oxidative stress, including that induced by UVA [32,33]. Our research supports the data, showing that UVA inhibits expression of ERK1/2 (Figure 4) and TGF-β (Figure 5A) to about 15% of the control, while AmO at a concentration of 0.1% and 0.15% partially restored the expressions to about 50% and 75% of the control and also restored to the control value a UVA-induced increase in the expression of p38 kinase (Figure 5B). This confirms the protective effect of AmO on the deleterious impact of UVA on the investigated cellular signaling pathways. Of interest is, however, the significant stimulatory effect of AmO on TGF-β expression. At a concentration of 0.15%, AmO stimulated TGF-β expression by about 2.5-fold. This finding requires further study on the mechanism of AmO-dependent stimulation of TGF-β expression.

UVA-induced skin damage is preceded by tissue inflammation. Nuclear factor-kappa B (NF-κB) has been described as one of the pro-inflammatory factors activated by UV radiation. It serves as a mediator of cellular responses to inflammatory stimuli, pathogens, and cellular stressors. In the cytoplasm of cells, NF-κB is present in an inactive form bound to the inhibitory protein IκB. It is known that NF-κB activation is induced by inflammatory stimuli, and subsequently, NF-κB translocates to the cell nucleus [34]. Activated NF-κB plays a crucial role in regulating the expression of various genes involved in immune and inflammatory responses, including the expression of cyclooxygenase COX-2, an enzyme participating in the inflammatory process. UVA radiation activates COX-2, which induces the inflammatory process. Vostálová et al. [12] observed an increase in the expression of COX-2 in the skin of mice exposed to a dose of 20 J/cm^2^ of UVA. In our experiments, we demonstrated that UVA irradiation of fibroblasts at a dose of 10 J/cm^2^ significantly increased the expression of NF-κB and its translocation to the cell nucleus (Figure 6A), along with increased COX-2 expression (Figure 6B). This confirms the pro-inflammatory mechanism of UVA in the studied cellular model. AmO by itself did not induce changes in the expression and translocation of NF-κB to the nucleus; however, in cells exposed to it, the UV radiation significantly inhibited the expression of this factor. These experiments show that a UVA-induced increase in the expression of COX-2 associated with the translocation of NF-κB into the nucleus of human skin fibroblasts is counteracted by AmO (Figure 6A,B). This shows protective action of AmO against UVA-induced inflammation in fibroblasts and suggests that AmO preparation may be useful as a potential therapeutic agent.

Disruption of skin integrity by external factors (e.g., UV radiation) activates reparative processes. Fibroblasts play a crucial role in skin wound healing, from the early inflammatory phase to the final production of extracellular matrix components. Contemporary research is exploring substances, including natural compounds, that can impact tissue regeneration. Plant oils have shown promising results in both in vitro and in vivo studies. They influence various phases of the wound healing process through their antimicrobial, anti-inflammatory, and antioxidant properties, as well as by promoting cell proliferation, enhancing collagen synthesis, stimulating skin regeneration, and repairing the skin’s lipid barrier function. It has been demonstrated that the fatty acids present in these oils also play a significant role in the wound healing process. Linolenic, linoleic, and oleic acids serve as precursors for the synthesis of inflammatory or anti-inflammatory mediators and are integral components of cell membrane phospholipids, ceramides, and sebum, all of which are vital constituents of the lipid barrier [35].

Plant-based oils are mixtures of glyceryl esters and higher fatty acids, and they also contain phytosterols, vitamin E and its derivatives, and squalene. It has been demonstrated that triglycerides, free fatty acids, glycerol, and non-saponifiable compounds directly or indirectly influence the wound healing process. The biological effect of plant oils in wound healing is mainly attributed to their similarity to skin lipids. Numerous studies confirm that various natural oils, under different conditions, exhibit more or less effective actions in the wound healing process. For example, in vitro studies have shown that cold-pressed rapeseed oil stimulates fibroblast proliferation and promotes fibroblast migration to the wound area, indicating its wound-healing effects. Similar effects were observed with flaxseed oil [35]. Additionally, plant oils like borage oil, evening primrose oil, and avocado oil, containing gamma-linolenic acid (GLA), are beneficial for skin conditions, inflammation, and irritations. Borage oil, with its high GLA content, stimulates skin cell activity and regeneration, making it useful in treating skin conditions such as allergies and inflammations. Evening primrose oil also supports skin regeneration, eases skin issues, and reduces inflammation, making it a good choice for those with conditions like psoriasis. Avocado oil, rich in vitamin E, β-carotene, vitamin D, protein, lecithin, and fatty acids, offers substantial benefits when added to formulations [36]. In respect to AmO, there is a lack of studies on its role in skin repair processes, so it was essential to conduct this study.

Wound healing involves well-defined repair stages, encompassing inflammation, proliferation, re-epithelialization, and remodeling. Various preclinical models, including those involving mice, rabbits, and pigs, offer the ability to simulate acute or impaired wounds, such as those associated with diabetes or nutritional deficiencies. These models can be induced through diverse techniques, with excision or incision being the most prevalent. Once a suitable model is identified for a particular study, researchers must carefully choose methods that are both appropriate and reproducible, enabling the systematic monitoring of wound progression over time. Evaluation methods can range from non-invasive protocols like wound tracing, photographic documentation (with subsequent image analysis), and biophysical techniques, to more invasive protocols requiring wound biopsies. The selection of these methodologies is crucial in ensuring accurate and comprehensive insights into the dynamics of wound healing under specific conditions [37]. Animal models and in vitro assays have evolved into essential tools for researchers across various scientific disciplines. This stepwise approach ensures a thorough understanding of the potential efficacy and safety of interventions before they progress to human trials, facilitating the development of more effective wound care strategies. In vitro studies are essential for assessing the potential efficacy of concentrations and evaluating the effectiveness of products on different cell types like fibroblasts and keratinocytes. They offer a cost-effective, rapid, and convenient approach for researchers. Notably, these assays provide valuable results swiftly and are ethically appealing as they typically do not involve the use of animals or humans. In the realm of wound healing research, in vitro assays are particularly valuable for assessing the potential effectiveness of various treatments, including antimicrobial and wound-healing enhancers [38]. The wound-healing assay is a straightforward and cost-effective technique that represents one of the earliest approaches for investigating directional cell migration in vitro. This method effectively replicates the cellular migration observed during the natural process of wound healing in vivo [39].

The results of wound closure are presented in images obtained from a fluorescence microscope and were subjected to an analysis using ImageJ^®^ 1.8.0 software with the Wound Healing Size Tool extension (Figure 7A,B). The wound healing closure rate was calculated and presented on a chart and in digital form (Figure 7C,D). The obtained results show that AmO stimulated fibroblast proliferation and promoted their migration in a model of wound healing under control conditions (without exposure to UVA radiation). All studied concentrations of AmO (0.05%, 0.1%, and 0.15%) positively influenced the wound healing process in a time- and dose-dependent manner. Interestingly, the ability of AmO to stimulate fibroblast proliferation and migration was also enhanced after exposure to UVA, although less intensely than in non-UVA-irradiated cells. The effectiveness of AmO in the wound healing process may be attributed to the presence of components such as linoleic acid, squalene, derivatives of vitamin E, and phytosterols. These components not only scavenge free radicals, as demonstrated in our previous study [13], but also have a beneficial impact on the wound healing processes, as presented in this report.

It is known that migration, adhesion, proliferation, neovascularization, remodeling, and apoptosis are the key processes in wound healing, affected by ROS. An increased production of ROS can disrupt these processes [40]. Studies by Wlaschek and Scharffetter-Kochanek [41] have shown that changes in mitochondrial DNA induced by UV radiation can increase intracellular ROS levels in fibroblasts, leading to alterations in their migration and proliferation. In vitro wound healing tests have demonstrated a significantly reduced ability of fibroblasts to close the wound gap, suggesting that an excess of ROS in mitochondria negatively affects the wound healing process. This explains how the overlapping processes, resulting in increased ROS production due to cell damage in fibroblasts exposed to UVA radiation, limit the regenerative capacity of the compound studied in our study, in the wound healing process [41].

Therefore, it is advisable, based on our research findings, to consider the use of unrefined plant-derived oils, including AmO, in cosmetics, with a protective effect against solar radiation. Therefore, results of this study provide rationale for applications of AmO in therapeutics of protective activity against UVA-induced skin damage.

It has been demonstrated that therapies involving the application of plant-derived butters or oils are efficacious with limited adverse effects, rendering phytotherapy a compelling alternative for dermatological treatments. Fatty acids are recognized to play a significant role in the described wound healing processes, whereas non-saponifiable compounds can substantially contribute to antimicrobial, antioxidant, and anti-inflammatory effects [28]; however, the application of oils in cellular studies faces several notable limitations. Firstly, the variable chemical composition of plant oils, influenced by factors such as the source, cultivation conditions, and extraction methods, can introduce fluctuations in research outcomes. Ensuring oil purity, especially in naturally obtained oils, poses a challenge, with impurities potentially yielding unpredictable effects in cell experiments. Additionally, the susceptibility of some plant oils to oxidation requires careful attention to stability and proper storage to preserve their biological properties. Interactions between oils and other components in a cell culture may influence experimental outcomes, necessitating consideration in data interpretation. Standardization becomes crucial due to variations in oil composition, aiding in the comparison of results across different experiments. Lastly, while plant oils often exhibit beneficial properties, their effects can be unpredictable depending on cell types and culture conditions. Awareness of these limitations is essential for researchers to adapt protocols effectively and ensure the reliability and reproducibility of their results.

Vegetable oils are distinguished by their chemical composition, which plays a pivotal role in determining their health properties. Analyzing characteristic chemical compositions, including fatty acid composition, tocols, phytosterols, squalene, carotenoids, phenolics, and phospholipids, provides insights into the diverse health benefits associated with various vegetable oils. These benefits encompass antioxidant activity, prevention of cardiovascular disease (CVD), anti-inflammatory, anti-obesity, and anti-cancer properties, diabetes treatment, and kidney and liver protection. The exploration of health benefits is rooted in the predominant components found in representative oils. Each type of vegetable oil exhibits a unique chemical composition, with significant variations in key components, emphasizing the distinct advantages and health effects associated with different oils [42]. In this context, *Amaranthus cruentus* L. seed oil (AmO) stands out with a favorable chemical composition, featuring approximately 60% linoleic acid, 8–10% squalene, and tocopherols. Moving beyond health properties, the next crucial aspect of AmO’s advantages lies in its chemical compositions, characterized by physicochemical properties such as the melting point, flash point, density, total acid number, iodine value, and saponification number. Qualitative and quantitative chromatographic analyses of phenolic compounds, determination of total antioxidant content (TAC), and identification of fatty acids and squalene further contribute to unraveling the comprehensive profile of AmO [43]. Therefore, AmO attracts attention as a unique mixture of many biologically active substances.

## 3. Materials and Methods

### 3.1. Materials

#### 3.1.1. Plant Materials

The *Amaranthus cruentus* L. seed oil used in this research is a commercially available product sourced from Szarłat M. and W. Lenkiewicz s.j. in Zawady, Poland. This oil was obtained through an unrefined seed cold-pressing process from locally cultivated *Amaranthus cruentus* plants. The fatty acid composition of the oil was as follows: palmitic acid accounted for 17.5–19.2% of the oil, oleic acid for 17.5–20.5%, linoleic acid for 52–55%, and stearic acid for 4.5–5.5%. Additionally, the oil contained approximately 6–8% of squalene and 8–10 mg of vitamin E per 100 g. For detailed safety information regarding the investigated *Amaranthus cruentus* L. seed oil, refer to the comprehensive safety data sheet (SDS) provided by the manufacturer, the Szarłat company (Zawady, Poland). This document is available in the Appendix A.

#### 3.1.2. Cell Lines and Culture

Human dermal fibroblasts were obtained from the American Type Culture Collection (Manassas, VA, USA); maintained in Dulbecco’s Modified Eagle Medium (DMEM) (PANTM BIOTECH, Aiden Bach, Germany); supplemented with 10% Fetal Bovine Serum (FBS) (Gibco, Waltham, MA, USA), 50 U/mL of Penicillin (Pen) (Gibco, Waltham, MA, USA), and 50 μg/mL of Streptomycin (Strep) (Gibco, Waltham, MA, USA); and incubated at 37 °C in 5% CO_2_. The cells were grown on 100 mm dishes in 10 mL of the complete medium. The cell culture medium was changed 2–3 times per week. The cells were used after 8–10 passages. For the experiments with the investigated oil, we used DMEM without FBS and Pen/Strep. The AmO was used in the following concentrations: 0.05%, 0.1%, 0.15%. After 30 min of incubation, the cell culture media containing the studied oil were removed and the plates were washed with PBS. Subsequently, the cells were exposed to UVA radiation using a Bio-Link Crosslinker BLX 365 (Vilber Lourmat, Eberhardzell, Germany) at a dose of 10 J/cm^2^ in 1 mL of cold PBS (4 °C). After irradiation, PBS was exchanged for fresh DMEM. The cells were incubated for 24 h. The concentration of AmO and the 10 J/cm^2^ dose of UVA were selected based on cell viability measurements obtained from the MTT assay conducted in our previous study (data shown in previous study) [13].

### 3.2. Methods

#### 3.2.1. Collagen Biosynthesis Assay

In this experiment, cells were cultured in 100 mm plates, with approximately 1 × 10^6^ cells per plate. The confluent fibroblast cells were treated as described in Section 3.1.2. Collagen biosynthesis in this research was evaluated using radioactive 5-[3H]-proline (5 μCi/mL; Hartmann Analytic, Braunschweig, Germany) incorporated into proteins that are susceptible to bacterial collagenase. Peterkofsky’s method [44] was employed for this purpose. After a 24 h incubation, the cells were washed with PBS (pH 7.4), collected in PBS containing 10 mM proline, and then stored at −80 °C until the day of the analysis. To perform collagen digestion, purified *Clostridium histolyticum* collagenase from Sigma Aldrich, Saint Louis, MO, USA, was used. A radiometric analysis was carried out using the Liquid Scintillation Analyzer Tri-Carb 2810 TR from PerkinElmer, Waltham, MA, USA. The results obtained from this analysis were standardized against total protein biosynthesis and presented as a percentage of the control value, providing valuable insights into collagen biosynthesis in the context of the study. Detailed data provided in Appendix A.

#### 3.2.2. Prolidase Activity

In this experiment, cells were cultured in 100 mm plates, with approximately 1 × 10^6^ cells per plate. The confluent fibroblast cells were treated as described in Section 3.1.2. After the 24 h incubation, the cells were collected and subjected to a prolidase assay, following the method described by Myara [45]. To determine the total protein concentration in the samples, the Lowry method [46] was used. Prolidase activity was reported as the nanomoles of proline released from the synthetic substrate (Gly-Pro) within one minute, normalized per milligram of the supernatant protein in the cell homogenate. Detailed data provided in Appendix A.

#### 3.2.3. Western Immunoblot Analysis and Quantifications of Western Blots with ImageJ^®^ 1.8.0 Software

For the analysis of the protein expression via Western blotting, the cells were cultured in 100 mm plates at about 2.0 × 10^6^ cells. The confluent fibroblast cells were treated as described in Section 3.1.2. After 24 h of incubation, the culture media were removed and the cells were harvested using a cell lysis buffer supplemented with a protease/phosphatase inhibitor cocktail. The protein concentrations of the samples were determined via the Lowry method [46]. Then, the proteins were separated using the SDS-PAGE method described by Laemmli [47]. After this step, the gels were washed in a cold Towbin buffer (25 mM Tris, 192 mM glycine, 20% (*v*/*v*) methanol, 0.025–0.1% SDS, pH 8.3). The proteins in the gels were transferred onto the 0.2 µm nitrocellulose membranes using a Trans-Blot (BioRad, Hercules, CA, USA). The transfer conditions were 200 mA and 3 h in a freshly prepared Towbin buffer and the temperature was maintained around 4–8 °C. The blocking of the membranes was performed using 5% NFDM for 1 h at RT. When the blocking was complete, the membranes were washed three times with 20 mL of TBS-T (20 mM Tris, 150 mM NaCl, and 0.1% Tween^®^ 20). After the washing step, the membranes were incubated with primary antibodies overnight at 4 °C. The concentration of the primary antibodies was 1:1000. Furthermore, the membranes were washed three times with 20 mL of TBS-T and a secondary antibody conjugated with HRP solutions (1:3000) in 5% NFDM that was used for 1 h at RT. Then, the membranes were washed 3 times with 20 mL of TBS-T and visualized using BioSpectrum Imaging System UVP (Ultra-Violet Products Ltd., Cambridge, UK).

Quantifications of Western blots with ImageJ^®^ 1.8.0 enable the relative quantification of protein bands on Western blot films without providing absolute values. The images were opened using the ImageJ program and converted to grayscale images by selecting Image > Type > 8-bit. Subsequently, the “Rectangular Selections” tool from the toolbar was chosen, and a rectangle was drawn around the initial band. Analyze > Gels > Select First Lane was then utilized to establish the selection for the first band. Clicking and holding in the middle of the rectangle on the first band, it was dragged over to the next band, and Analyze > Gels > Select Next Lane was used to set the parameters for the second band. The process continued until all the bands were selected. For a further analysis, Analyze > Gels > Plot Lanes was chosen to create profile plots for each lane. Using the “Straight Line” selection tool from the toolbar, lines across the base of the peaks in the profile plot to enclose each peak were drawn. The “Wand tracing” tool from the ImageJ toolbar was selected and clicked on inside each peak, and the area of each peak was shown in a new window. This step was repeated until peak areas for all bands were collected [48,49,50]. Detailed data provided in Appendix A.

#### 3.2.4. Immunofluorescence Staining and Confocal Microscopy

The cells were cultured on a black 96-well plate. The confluent fibroblast cells were treated as described in Section 3.1.2. After 24 h, the culture media were removed and the cells were fixed with a 3.7% formaldehyde solution at room temperature for 10 min. Then, the plate was washed once with 100 µL/well of PBS. Later, the permeabilization with a 0.1% Triton X-100 solution and a 10 min step was performed. After the permeabilization, the plate was washed twice with PBS and 3% FBS was used as a blocking agent at room temperature for 30 min. After the FBS removal, 50 µL of the primary antibody (1:50), diluted in 3% FBS, was added and the plate was incubated for 1 h at room temperature. After the incubation, the primary antibody plate was washed three times with PBS. Then, 50 µL per well of the secondary antibody (dilution 1:1000) was added for the next 1 h. During this step, the plate was covered from the light. When the secondary antibody solution was removed, the plate was washed 3 times with PBS and the wells were filled with 100 µL of PBS containing 2 µg/mL of Hoechst 33342 for the nuclei staining. The plate was visualized using a confocal laser scanning microscope, BD Pathway 855 (Bioimager, Becton Dickinson, Franklin Lakes, NJ, USA), supported with AttoVision^TM^ 1.6 software.

#### 3.2.5. Cell Migration Assay

The wound-healing assay is a straightforward and cost-effective technique that represents one of the earliest approaches for investigating directional cell migration in vitro. This method effectively replicates the cellular migration observed during the natural process of wound healing in vivo. The fundamental procedure entails inducing a simulated “wound” within a cell monolayer, documenting images at the initiation and at predetermined intervals throughout the migration process, and subsequently analyzing the images to quantify the rate of cell migration [39]. The key information obtained from the wound-healing assay typically centers on the rate of gap closure, acting as a measure for the speed of collective cell movement. A cell-free gap within a cell monolayer can be generated through either direct manipulation or physical exclusion. To mimic the wound, the prevalent method involves inducing a gap by gently scratching a fully confluent monolayer using a pipette tip [51].

Confluent fibroblasts plated in 6-well plates were scratched with a sterile 200 µL pipette tip and treated as described in Section 3.1.2. The scratched area was monitored with an inverted optical microscope at 40× magnification (Nikon; Minato, Tokyo, Japan). Fibroblast migration was calculated using the dedicated Wound Healing tool for ImageJ^®^ 1.8.0 software and its rate was calculated according to the following formula:$$\mathrm{wound}\;\mathrm{healing}\;\mathrm{rate}=\frac{original\;wound\;area-unhealed\;wound\;area}{original\;wound\;area}$$

#### 3.2.6. Statistical Analysis

The mean ± standard error (SEM) values from the experiments performed in triplicates are presented for collagen biosynthesis and prolidase activity. Student’s *t*-test was employed to assess statistical differences between the tested parameters. Statistically significant differences are marked as * according to the control and ^##^ according to only the UVA-treated cells at *p* < 0.05 and described in the legend to figures. All graphs were created using the Microsoft Excel program, and the standard error (SEM) was automatically generated in the Excel tool for the graphs. Plot bars, plot measurements for Western blot densitometry, and wound healing images were generated using ImageJ^®^ 1.8.0 software.

## 4. Conclusions

UVA radiation impairs collagen biosynthesis, prolidase activity, and the expression of the β1-integrin receptor, IGF receptor, phosphorylated forms of ERK 1/2 kinases, and TGF-β factor, while it stimulates the expression of p38 kinase. The application of AmO reversed the adverse effects induced by UVA on these processes. UVA radiation induces the expression and nuclear translocation of the pro-inflammatory NF-κB factor and enhances COX-2 expression. The use of AmO significantly inhibited the expression of these factors. *Amaranthus cruentus* L. seed oil stimulates the proliferation and migration of fibroblasts in a wound healing model. After cell exposure to ultraviolet radiation, AmO promoted the wound healing process. The results of these studies allow us to assume that the presented properties of AmO may be used in preparations that protect the skin against the harmful effects of UV radiation.

## Figures and Tables

**Figure 1 ijms-25-00925-f001:**
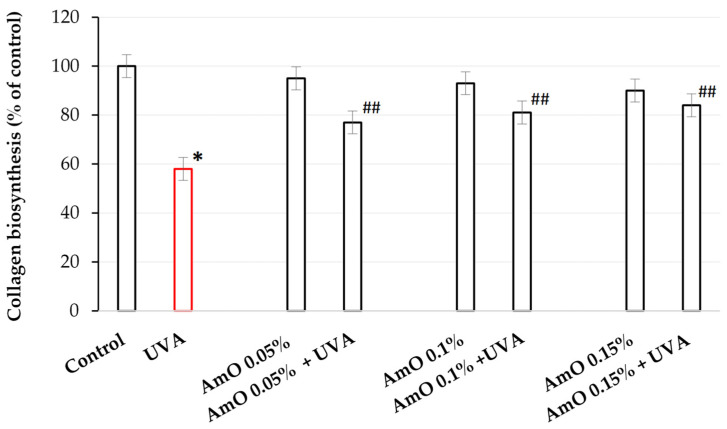
Collagen biosynthesis in fibroblasts irradiated with UVA and treated with AmO at concentrations of 0.05%, 0.1%, and 0.15% vs. the control. The mean ± standard error (SEM) values from the experiments performed in triplicates. * Statistically significant differences at *p* < 0.05 compared with the control. **^##^** Statistically significant difference at *p* < 0.05 compared to UVA (red bar).

**Figure 2 ijms-25-00925-f002:**
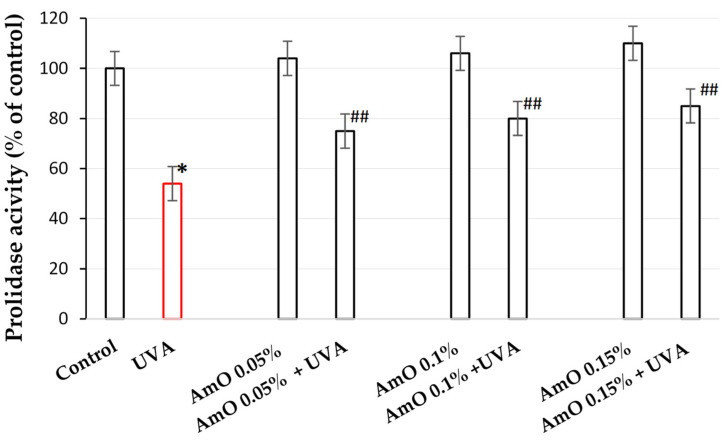
Prolidase activity in fibroblasts irradiated with UVA and treated with AmO at concentrations of 0.05%, 0.1%, and 0.15% vs. the control. The mean ± standard error (SEM) values from the experiments performed in triplicates. * Statistically significant differences at *p* < 0.05 compared with the control. ^##^ Statistically significant difference at *p* < 0.05 compared to UVA (red bar).

**Figure 3 ijms-25-00925-f003:**
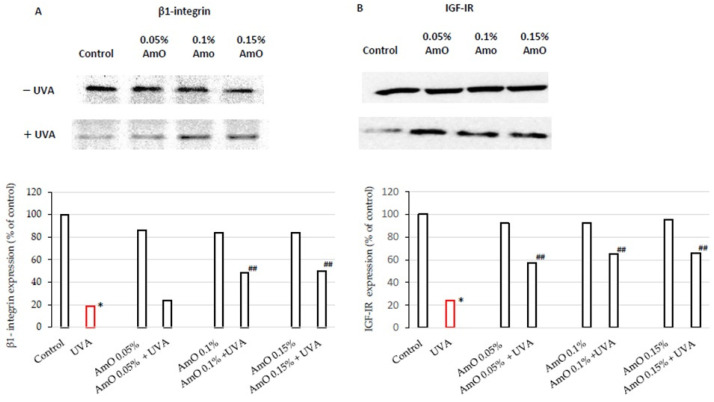
Western blot and densitometry (completed using ImageJ^®^ 1.8.0) for β1-integrin receptor (**A**) and IGF-IR (**B**) from 3 pooled homogenates of UVA-irradiated fibroblasts cultured in the presence of AmO at concentrations of 0.05%, 0.1%, and 0.15% vs. the control. * Statistically significant differences at *p* < 0.05 compared with the control. ^##^ Statistically significant difference at *p* < 0.05 compared to UVA (red bar).

**Figure 4 ijms-25-00925-f004:**
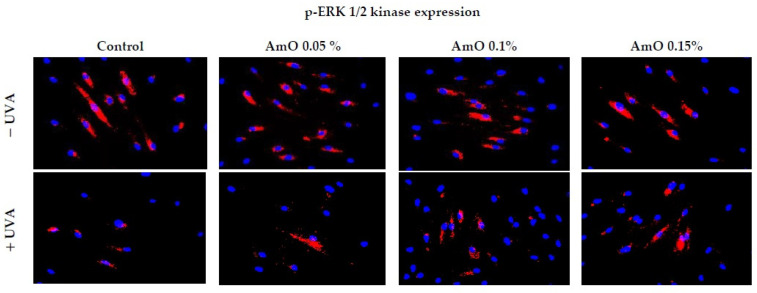
Expression of p-ERK 1/2 kinase (visualized using immunofluorescence staining) in fibroblasts irradiated by UVA and treated with AmO at concentrations of 0.05%, 0.1%, and 0.15% vs. the control. Blue staining indicates the nuclei and red staining represents ERK 1/2 expression. The images were obtained at a 20× magnification.

**Figure 5 ijms-25-00925-f005:**
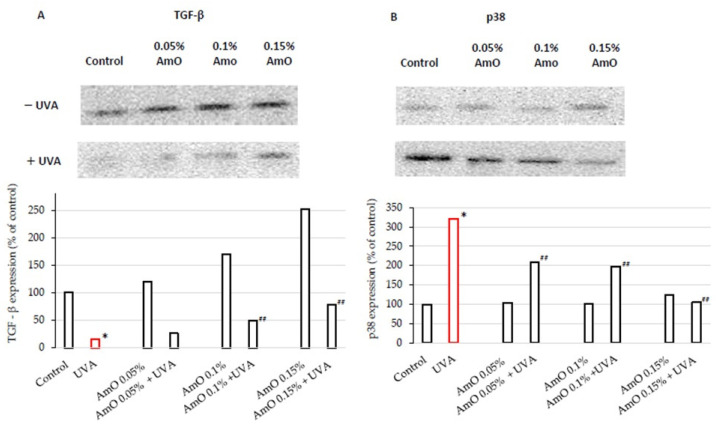
Western blot and densitometry (completed using ImageJ^®^ 1.8.0) for TGF-β (**A**) and p38 (**B**) from 3 pooled homogenates of UVA-irradiated fibroblasts cultured in the presence of AmO at concentrations of 0.05%, 0.1%, and 0.15% vs. the control. * Statistically significant differences at *p* < 0.05 compared with the control. ^##^ Statistically significant difference at *p* < 0.05 compared to UVA (red bar).

**Figure 6 ijms-25-00925-f006:**
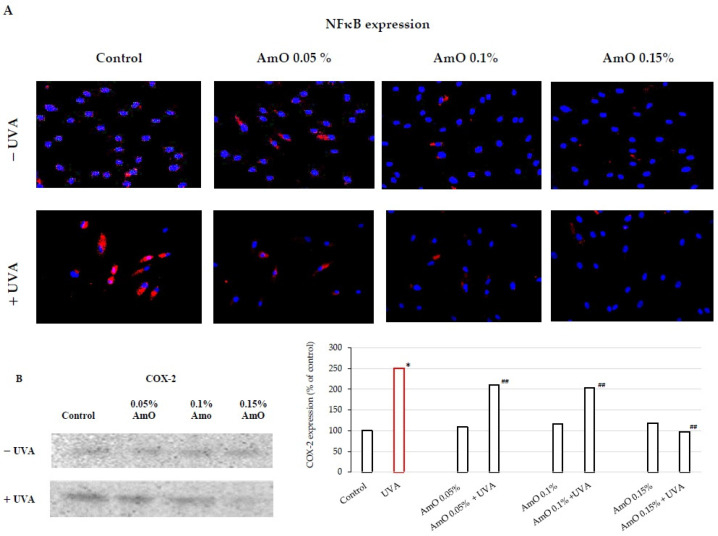
(**A**) Expression of NFκB (visualized using immunofluorescence staining) in fibroblasts irradiated by UVA and treated with AmO at concentrations of 0.05%, 0.1%, and 0.15% vs. the control. Blue staining indicates the nuclei and red staining represents NF-κB expression. The images were obtained at a 20× magnification. (**B**) Western blot and densitometry (completed using ImageJ^®^ 1.8.0) for COX-2 from 3 pooled homogenates of UVA-irradiated fibroblasts cultured in the presence of AmO at concentrations of 0.05%, 0.1%, and 0.15% vs. the control. * Statistically significant differences at *p* < 0.05 compared with the control. ^##^ Statistically significant difference at *p* < 0.05 compared to UVA (red bar).

**Figure 7 ijms-25-00925-f007:**
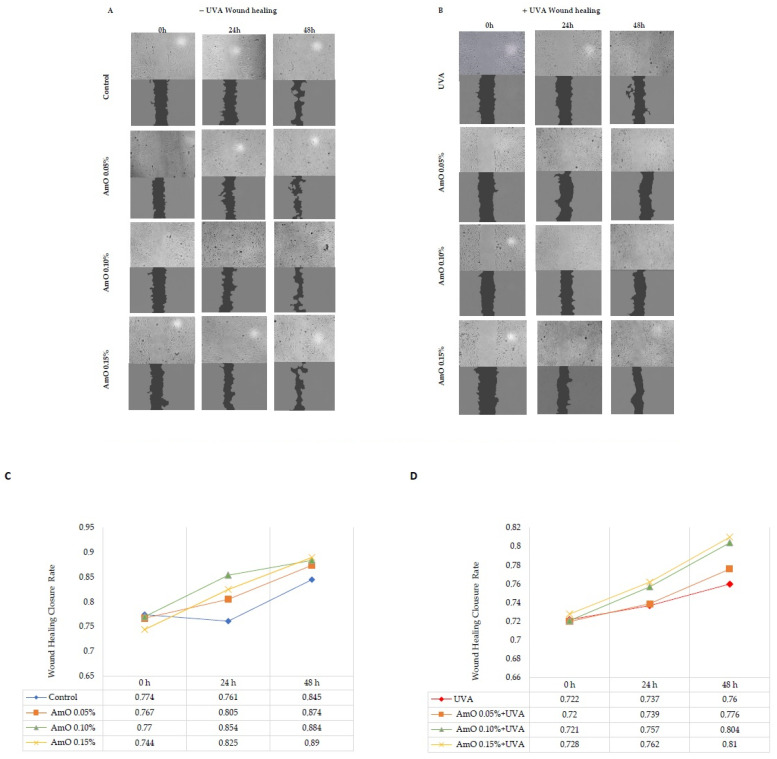
Wound healing images in (**A**) non-UVA-irradiated fibroblasts, treated for 24 h and 48 h with AmO at concentrations of 0.05%, 0.1%, and 0.15%, and (**B**) UVA-irradiated fibroblasts, treated for 24 h and 48 h with AmO at concentrations of 0.05%, 0.1%, and 0.15%. The wound healing closure rate was calculated and analyzed using ImageJ^®^1.8.0 software in (**C**) non-UVA-irradiated fibroblasts, treated for 24 h and 48 h with AmO at concentrations of 0.05%, 0.1%, and 0.15%, and (**D**) UVA-irradiated fibroblasts, treated for 24 h and 48 h with AmO at concentrations of 0.05%, 0.1%, and 0.15%.

## Data Availability

The data presented in this study are available in the main text of this article or on request from the corresponding author.

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
