# Peer review of "Amaranthus cruentus L. Seed Oil Counteracts UVA-Radiation-Induced Inhibition of Collagen Biosynthesis and Wound Healing in Human Skin Fibroblasts"

_ijms, 2024, doi:10.3390/ijms25020925_

Round 1
Reviewer 1 Report
Comments and Suggestions for Authors
The paper by Wolosik et al describes the protective effect of Amaranthus cruentus seed oil on the dentrimental effect of UVA-radiation in human skin. It is an interesting paper, however I have some points that need to be addressed.
Major:
In all figures resulted from images (such as westerns and Immunofluoresensce ) the statistics are missing. What is the n in these experiments?Bar graphs and quantification means that the authors have performed statistics also.
In all figure legends the number of repeats in each experiment is missing, as well as the type of statistics.
Page4 line 138 and Figure 3. It seems to me that the lower dose of AmO does have an effect on b1 integrin and not the higher dose. Furthermore, how did the authors stated that the effect is dose-dependent since there are no error bars?
Furthermore, in all treatments and experiments, it seems to me that the dose 0.15% (higher one ) is the least effective. How can the authors explain this phenomenon?
Did the authors try to treat the cells with the seed oil after the UVA irradiation? If yes what is the effect?
Finally, I can not open the images (not published data). It seems to me that they are not loaded successfully.
Author Response
We agree with all comments raised by the Reviewer. According to the Reviewer’s suggestion, we made the following changes (red labeled sentences) in the revised manuscript:
Answer to the Reviewer 1
The paper by Wolosik et al describes the protective effect of Amaranthus cruentus seed oil on the detrimental effect of UVA-radiation in human skin. It is an interesting paper, however I have some points that need to be addressed.
Major:
In all figures resulted from images (such as Westerns and Immunofluorescence) the statistics are missing. What is the n in these experiments? Bar graphs and quantification means that the authors have performed statistics also.
In all figure legends the number of repeats in each experiment is missing, as well as the type of statistics.
Page 4 line 138 and Figure 3. It seems to me that the lower dose of AmO does have an effect on b1 integrin and not the higher dose. Furthermore, how did the authors stated that the effect is dose-dependent since there are no error bars?
Furthermore, in all treatments and experiments, it seems to me that the dose 0.15% (higher one ) is the least effective. How can the authors explain this phenomenon?
We extend our sincere apologies for any confusion caused by the interpretation of results, particularly in Figures 3, 5, and 6, related to western blot analyses. Regrettably, an oversight led to the inclusion of analyses in the article where unexposed groups were compared to controls, while exposed groups were compared to the UVA-only group. This unintentional error resulted in an inaccurate interpretation of the results, rendering the accompanying captions inadequate for these figures.
To address this issue, the authors have conducted a new gel plot analysis of the densitometry results using ImageJ software and the gel plot analysis tool. We are attaching the correct results, which should now provide a clear and accurate representation of the findings described in the text of the manuscript.
Additionally, we have included the outcomes of bar measurements and images of the western blot gel densitometry bars obtained from ImageJ software for enhanced visualization of the results. These supplementary materials are available in Supplementary Materials 2.
Once again, we appreciate the diligence of the reviewers in bringing this matter to our attention, and we are confident that the corrected analyses will rectify any misinterpretation.
Did the authors try to treat the cells with the seed oil after the UVA irradiation? If yes what is the effect?
We did not attempt to treat the cells with the seed oil after UVA irradiation in this study. However, we express the intention to consider this procedure in similar research in the future.
Finally, I can not open the images (not published data). It seems to me that they are not loaded successfully.
We reloaded unpublished data
We thank the Reviewer for all these insightful suggestions.

Reviewer 2 Report
Comments and Suggestions for Authors
Manuscript ID: ijms-2787783 – Amaranthus cruentus L. Seed Oil counteracts UVA - Radiation –induced inhibition of Collagen Biosynthesis and Wound Healing in Human Skin Fibroblasts
Reviewer Comments
Overall
The authors report the results of several in vitro studies on the effects of Amaranthus cruentus L. seed oil on fibroblasts exposed to UVA radiation. They assessed the effects of three concentrations on mechanisms involved with collagen biosynthesis, known to be impacted by the inflammatory cascade associated with UVA. Given the detrimental effects of UVA exposure, identification of modalities to mitigate the effects is important and, in general, applies to a variety of wounds.
The following comments are intended to clarify the paper and address potential questions from the readers.
Introduction
1. Please elaborate on the statement beginning in line 57 beginning “Its use may contribute to various health benefits”. The statement is not specific. For example, further detail is needed to indicate how it might serve as an anti-allergic and anti-hypertensive.
2. Some of the information in lines 70-94 is more suited for the introduction, including the author’s previous work on cell viability. Provide additional information on that study to help explain why the current studies were conducted.
3. Explain the uniqueness or difference in AmO compared to other oils with high levels of unsaturation and squalene, for example. What makes this oil of interest?
Materials and Methods
1. In line 310, what is meant by “validated” commercially available product? Also, please comment about the variability in the AmO composition from batch to batch as this will be important for generalizing the findings.
2. For the Cell Migration Assay, add more information about this method, including citation of a previous work showing it is a valid method for wound healing.
3. In all sections, add more detail on the specific details of how ImageJ was used and what specific parameters from images were quantified. Consider providing some image analysis data as a supplementary file.
4. Provide a section in the methods about the statistical analysis procedures used in the various experiments reported in the paper.
5. Provide details about how the UVA radiation was delivered and the specific dose that was used. Did the authors consider varying the UVA dose and why or why not? How does the UVA dose compare to typical UVA exposure used for the evaluation of cosmetic products or OTC sunscreens.
Results and Discussion
1. Did the authors consider using a known UVA protectant (one used in common sunscreens) as a control? Why or why not?
2. For all analyses, beginning with the collagen biosynthesis report in Figure 1, did the authors compare the AmO with UVA treatment (all 3 levels) with the UVA treated control. How did each oil plus UVA compare to both the control and UVA alone? If the experiments were conducted at the same time, would it not be appropriate to analyze the data this way?
3. In most of the figures, there does not appear to be a dose effect for 0.05%, 0.10% and 0.15% AmO? In fact, it appears that the higher oil level is “less effective” but the statistical analysis has to be shown.
4. For Figures 3A and B, were there statistical analyses applied to the data? Pleaser report them.
5. What is the explanation in figure 3B for the decrease for AmO at 0.05% and the increase or no difference for the other two levels?
6. Line 187 states that there was a dose-dependent effect for p-ERK ½ and TGF-b. Please show the statistics and explain.
7. Show the statistics for the data in Figure 6.
8. Explain how the results of the wound healing evaluation would likely predict the results for traditional methods of wound healing, such as in vivo wound healing models. Said differently, does a positive finding with this method correlate to animal models of wound healing, etc.?
9. Add a section to the discussion section regarding the generalizability and also about the limitations.
10. Discuss next steps so readers can understand what needs to be done to evaluate the effects in vivo and/or in vitro systems that model full thickness skin.
11. Because AmO is a complex mixture, how would one determine the ingredients (or combination of ingredients) responsible for the observed effects?
Author Response
We agree with all comments raised by the Reviewers. According to the Reviewer’s suggestion, we made the following changes (red labelled sentences) in the revised manuscript:
Answer for the Reviewer 2
The following comments are intended to clarify the paper and address potential questions from the readers.
Introduction
- Please elaborate on the statement beginning in line 57 beginning “Its use may contribute to various health benefits”. The statement is not specific. For example, further detail is needed to indicate how it might serve as an anti-allergic and anti-hypertensive.
- Some of the information in lines 70-94 is more suited for the introduction, including the author’s previous work on cell viability. Provide additional information on that study to help explain why the current studies were conducted.
- Explain the uniqueness or difference in AmO compared to other oils with high levels of unsaturation and squalene, for example. What makes this oil of interest?
The above comments has been addressed and the text has been improved in lines 55-137
Seed oil derived from Amaranthus cruentus L. (AmO) is considered a promising source of bioactive compounds counteracting UVA-induced skin damage [13]. The grains of Amaranthus cruentus L. cultivated for their seeds exhibit a greater lipid content (7.2%) compared to traditional cereal plants. The lipid content is more than that found in wheat (1.9%), rye (1.9%) , or rice (2.1%) grains, and maize (4.5%). The seeds contain crude fat ranging from 5.6% to as much as 10.9%. Not only is the quantity favorable, but also the composition of lipids in amaranth grain is advantageous [14]. The oil derived from Amaranthus cruentus L. has been analyzed by León-Camacho et al. [15] to identify both major and minor compounds. Fatty acids, triglycerides, sterols, methyl sterols, terpenic and aliphatic alcohols, tocopherols, and hydrocarbons have been meticulously identified using standards and mass spectrometry. The content of these compounds, along with the equivalent carbon numbers (ECN) and triglyceride carbon numbers (TCN), has been compared with the findings of other edible vegetable and cereal oils such as canola, coconut, cottonseed maize, palm, soya, sunflower and others. The hydrocarbon composition is particularly noteworthy, with a predominant presence of squalene (4.16 g/kg of seed) and a concentration of n-alkenes (C23:1–C33:1) reaching 332 ppm (part per million) [15].
The oil composition is predominantly characterized by unsaturated fatty acids, specifically oleic acid, linoleic acid, and linolenic acid, which collectively contribute to diverse health-promoting effects. Certain unsaturated fatty acids, notably linolenic acid, are exogenous acids indispensable for the human body, serving a crucial role in hormone synthesis, cellular membrane structure, and the regulation of membrane permeability [14]. The findings in Martirosyan et al. [16] study indicate that Amaranthus oil has the potential to lower cholesterol levels in the bloodstream, making it a viable candidate for inclusion in functional food products designed for the prevention and treatment of cardiovascular diseases. Adopting a diet enriched with Amaranthus oil may contribute to lowering blood pressure, presenting itself as a promising alternative to pharmacological interventions for individuals with hypertension. Notably, the study suggests that the synergistic combination of Amaranthus oil with a hypo sodium antiatherogenic diet is more effective in reducing blood cholesterol levels compared to solely the hypo sodium antiatherogenic diet. These insights emphasize the potential of incorporating Amaranthus oil into dietary strategies for cardiovascular health [16].
Due to its abundant nutritional properties, certain Amaranthus preparations have found applications in the cosmetics industry for hair and skin health providing exceptional moisturizing capabilities, soothing irritations, promoting accelerated wound healing, and demonstrating antimicrobial properties. Notably, Amaranthus seed oil contributes to skin nourishment and exhibits anti-aging effects, facilitating the regeneration, nourishment, and strengthening of the epidermis. Functioning as an antioxidant, it offers protective qualities [17]. Lacatusu and colleagues explored innovative sunscreen formulations incorporating nanostructured lipid carriers (NLCs) as delivery systems for antioxidant and anti-UV bioactive. In this study, Amaranthus oil, along with pumpkin seed oil, was integrated into the lipid NLCs core, creating novel delivery systems capable of simultaneously encapsulating UVA and UVB filters and an antioxidant. This pioneering approach represents a non-invasive design for herbal cosmetic formulations, delivering superior photoprotection and enhanced antioxidant properties [17, 18].
Amaranthus cruentus L. seed oil contains substantial amounts of squalene. Depending on the variety, the oil extracted from these seeds can contain up to 8% of squalene. As a natural component of human skin lipids, squalene plays a crucial role in skin physiology, constituting approximately 13% of human sebum [19]. Squalene is characterized by high biological activity; it serves as a potent antioxidant, a natural emollient, and also reduces water loss from the epidermis, thereby influencing skin hydration. This property is utilized in preparations for treating skin conditions such as atopic dermatitis or psoriasis. Squalene also restricts the development of acne lesions in acne vulgaris [19]. In contemporary biomedical and cosmetic applications, squalene and its hydrogenated derivative, squalane, are employed in pharmaceutical and dermatological formulations for addressing skin disorders, leveraging their emollient and moisturizing properties. Furthermore, squalene has been incorporated into antiviral emulsions, utilized as an adjuvant in vaccines, and is traditionally employed in China as agents with anti-fatigue and anti-aging attributes [15].
The daily intake of squalene among individuals in Mediterranean countries can vary from 30 mg to 200 mg and even up to 1000 mg. The recommended minimum daily intake is 11 mg [19, 20]. Therapeutic doses of squalene, ranging from 2000 to 5000 mg per day, are considered effective against cancer. This leads to a reduced risk of bowel, breast, and skin cancer [19, 21]. Squalene demonstrates the ability to inhibit tumor cell growth and induce regression in existing tumors [17, 22, 23, 24]. In recent years, the protective role of squalene has been demonstrated against various carcinogens, including those associated with conditions like cell leukemia [24] or skin cancer [25]. Squalene also enhances the effects of antineoplastic agents such as adriamycin, 5-fluorouracil, or bleomycin [26]. In addition to alkylglycerols and omega-3s, squalene exerts a beneficial effect on the natural immune system and can be helpful in the treatment of diseases associated with impaired immune function [19, 26, 27].
We have found that in AmO at 0.05%, 0.1%, and 0.15% concentrations evoked protective effect on decreased viability of fibroblasts exposure to UVA irradiation [13]. Exposure to UVA radiation at a dosage of 10 J/cm2 resulted in the downregulation of p-AKT and mTOR expression, alongside the upregulation of p53, caspase-3, caspase-9, and PARP expression. This led to a substantial reduction in the viability of human skin fibroblasts, triggering apoptosis. Treatment of UVA-exposed fibroblasts with Amaranthus cruentus seed oil (AmO) at concentrations of 0.1% and 0.15% mitigated the expression of apoptosis-related markers (p53, caspase-3, caspase-9, and PARP) and reinstated the expression of p-Akt and mTOR proteins. The underlying molecular mechanism of this phenomenon is linked to the stimulation of antioxidant processes through the activation of Nrf2. This implies that AmO stimulates the antioxidant system in fibroblast cells, counteracting the effects of UVA-induced oxidative stress [13]. These data led us to evaluate the effect of AmO on collagen metabolism in fibroblasts exposed to UVA radiation.
Materials and Methods
- In line 310, what is meant by “validated” commercially available product? Also, please comment about the variability in the AmO composition from batch to batch as this will be important for generalizing the findings.
The term "validated" in line 310, means that the provided specifications by the oil producer company are established based on internal company standards and adhere to current cosmetics regulations. The term "validated" indicates that the product undergoes periodic testing once a year, following Annex III of Regulation 1223/2009/EC dated November 30, 2009, on cosmetic products. Furthermore, we have attached the Material Safety Data Sheet in the Supplementary Materials to provide transparency regarding the purity and composition of the tested Amaranthus cruentus seed oil. We hope this additional information enhances the clarity and solidity of our study.
To avoid confusion among readers, the sentence in line 484 has been changed to: “The Amaranthus cruentus L. seed oil used in this research is a commercially available product sourced from SzarÅ‚at M. and W. Lenkiewicz s.j. in Zawady, Poland”.
Regarding the potential variability in the composition of Amaranthus cruentus seed oil (AmO) from batch to batch, we acknowledge the importance of this aspect for generalizing the study findings. Although not explicitly addressed in the manuscript, we appreciate Your insight into the potential impact of natural variations. Natural products often exhibit composition differences due to factors such as plant genetics, growing conditions, and harvesting methods. Should there be significant variability in AmO composition between batches, we recognize the potential influence on the reproducibility and generalizability of our study results. We will consider and address this variability in our next similar experimental design and data interpretation.
- For the Cell Migration Assay, add more information about this method, including citation of a previous work showing it is a valid method for wound healing.
Informations were added in lines 358-384 and in 4.2.6. Cell Migration Assay section
Wound healing involves well defined repair stages, encompassing inflammation, proliferation, reepithelialization, and remodeling. Various preclinical models, including those involving mice, rabbits, and pigs, offer the ability to simulate acute or impaired wounds, such as those associated with diabetes or nutritional deficiencies. These models can be induced through diverse techniques, with excision or incision being the most prevalent. Once a suitable model is identified for a particular study, researchers must carefully choose methods that are both appropriate and reproducible, enabling the systematic monitoring of wound progression over time. Evaluation methods can range from non-invasive protocols like wound tracing, photographic documentation (with subsequent image analysis), and biophysical techniques, to more invasive protocols requiring wound biopsies. The selection of these methodologies is crucial in ensuring accurate and comprehensive insights into the dynamics of wound healing under specific conditions. [37]. Animal models and in vitro assays have evolved into essential tools for researchers across various scientific disciplines. This stepwise approach ensures a thorough understanding of the potential efficacy and safety of interventions before they progress to human trials, facilitating the development of more effective wound care strategies. In vitro studies are essential for assessing the potential efficacy of concentrations and evaluating the effectiveness of products on different cell types like fibroblasts and keratinocytes. They offer a cost-effective, rapid, and convenient approach for researchers. Notably, these assays provide valuable results swiftly and are ethically appealing as they typically do not involve the use of animals or humans. In the realm of wound healing research, in-vitro assays are particularly valuable for assessing the potential effectiveness of various treatments, including antimicrobial and wound-healing enhancers [38]. The wound-healing assay is a straightforward and cost-effective technique that represents one of the earliest approaches for investigating directional cell migration in vitro. This method effectively replicates the cellular migration observed during the natural process of wound healing in vivo [39].
- In all sections, add more detail on the specific details of how ImageJ was used and what specific parameters from images were quantified. Consider providing some image analysis data as a supplementary file.
Informations were added into sections 4.2.4 and 4.2.6. and Supplementary Materials.
Quantifications of Western Blots with ImageJ® software enable the relative quantification of protein bands on Western blot films without providing absolute values. The images were opened using the ImageJ program and converted to grayscale images by selecting Image > Type > 8-bit. Subsequently, the "Rectangular Selections" tool from the toolbar was chosen, and a rectangle was drawn around the initial band. Analyze > Gels > Select First Lane was then utilized to establish the selection for the first band. Clicking and holding in the middle of the rectangle on the first band, it was dragged over to the next band, and Analyze > Gels > Select Next Lane was used to set the parameters for the second band. The process continued until all the bands were selected. For further analysis, Analyze > Gels > Plot Lanes were chosen to create profile plots for each lane. Using the "Straight Line" selection tool from the toolbar, lines across the base of the peaks in the profile plot to enclose each peak were drawn. The "Wand tracing" tool from the ImageJ toolbar was selected and clicked inside each peak, and the area of each peak was shown in a new window. This step was repeated until peak areas for all bands were collected [48, 49, 50].
The wound-healing assay is a straightforward and cost-effective technique that represents one of the earliest approaches for investigating directional cell migration in vitro. This method effectively replicates the cellular migration observed during the natural process of wound healing in vivo. The fundamental procedure entails inducing a simulated "wound" within a cell monolayer, documenting images at the initiation and at predetermined intervals throughout the migration process, and subsequently analyzing the images to quantify the rate of cell migration [39]. The key information obtained from the wound healing assay typically centers on the rate of gap closure, acting as a measure for the speed of collective cell movement. A cell-free gap within a cell monolayer can be generated through either direct manipulation or physical exclusion. To mimic the wound, the prevalent method involves inducing a gap by gently scratching a fully confluent monolayer using a pipette tip [51].
- Provide a section in the methods about the statistical analysis procedures used in the various experiments reported in the paper.
Statistical analysis paragraph was added into 4.2.7 section
The mean ± standard error (SEM) values from the experiments performed in triplicates are presented for collagen biosynthesis and prolidase activity. The t-Student's test was employed to assess statistical differences between the tested parameters. Statistically significant differences are marked as * according to the control and ## according to only the UVA-treated cells at p < 0.05 and described in the legend to figures. All graphs were created using the Microsoft Excel program, and the standard error (SEM) was automatically generated in the Excel tool for the graphs. Plot bars, plot measurements for Western Blot densitometry, and wound healing images were generated by ImageJ® software.
- Provide details about how the UVA radiation was delivered and the specific dose that was used. Did the authors consider varying the UVA dose and why or why not? How does the UVA dose compare to typical UVA exposure used for the evaluation of cosmetic products or OTC sunscreens.
The biological effect of UV radiation depends on the wavelength of the radiation, the type of cells it interacts with, and the radiation dose. Excessive exposure to UV radiation can adversely impact cell survival. In studies on the influence of UVA radiation on skin cells, various doses of this radiation were applied, such as: 30 J/cm2 and 60 mJ/cm2 or 100, 60 and 20 J/cm2. Given the various UVA radiation doses reported in the literature, we conducted our previous study an assessment to determine the extent to which different doses of this radiation affect cells. Based on the results obtained, we observed, similar to other studies, that the viability of fibroblasts decreased with increasing UVA doses. The MTT test was employed and data are shown in our previous study. From the earlier conducted experiment, we selected a UVA dose of 10 J/cm2 for this experiment. Also, a dose of 10 J/cm2 corresponds to the daily dose to which the skin is subjected.
Results and Discussion
- Did the authors consider using a known UVA protectant (one used in common sunscreens) as a control? Why or why not?
The decision not to include known UVA protectant, as controls, was made due to the lack of established standards in the literature and the complexity of the compositions of various products. In the literature review, we did not find consistent guidelines regarding the use of specific protective substances as controls in studies similar to ours. Additionally, SPF creams vary in composition, containing different types of filters, both physical and chemical, adding an extra layer of complexity.
Certainly, we will consider incorporating a known UVA protectant, commonly found in sunscreens, as a control in future studies. Moving forward, we recognize the value of exploring the inclusion of UVA protective substances to enhance the comprehensiveness and robustness of our experimental design.
- For all analyses, beginning with the collagen biosynthesis report in Figure 1, did the authors compare the AmO with UVA treatment (all 3 levels) with the UVA treated control. How did each oil plus UVA compare to both the control and UVA alone? If the experiments were conducted at the same time, would it not be appropriate to analyze the data this way?
In our study, we simultaneously investigated the impact of the oil itself on various parameters in fibroblast cells and how the addition of the oil influenced them after UVA irradiation. We observed that the addition of selected concentrations of the oil did not affect negatively the examined parameters; however, it mainly had a positive influence on the results obtained after UVA irradiation. When discussing the results, we focused primarily on this aspect. We appreciate the suggestions from the reviewers, and the provided information has been appropriately incorporated into the manuscript in legends to Figures.
- In most of the figures, there does not appear to be a dose effect for 0.05%, 0.10% and 0.15% AmO? In fact, it appears that the higher oil level is “less effective” but the statistical analysis has to be shown.
- For Figures 3A and B, were there statistical analyses applied to the data? Please report them.
- What is the explanation in Figure 3B for the decrease for AmO at 0.05% and the increase or no difference for the other two levels?
- Line 187 states that there was a dose-dependent effect for p-ERK ½ and TGF-b. Please show the statistics and explain.
- Show the statistics for the data in Figure 6.
We extend our sincere apologies for any confusion caused by the interpretation of results, particularly in Figures 3, 5, and 6, related to western blot analyses. Regrettably, an oversight led to the inclusion of analyses in the article where unexposed groups were compared to controls, while exposed groups were compared to the UVA-only group. This unintentional error resulted in an inaccurate interpretation of the results, rendering the accompanying captions inadequate for these figures.
To address this issue, the authors have conducted a new gel plot analysis of the densitometry results using ImageJ software and the gel plot analysis tool. We are attaching the correct results, which should now provide a clear and accurate representation of the findings described in the text of the manuscript.
Additionally, we have included the outcomes of plot measurements and images of the western blot gel densitometry plots obtained from ImageJ software for enhanced visualization of the results. These supplementary materials are available in Supplementary Materials 2.
Once again, we appreciate the diligence of the reviewers in bringing this matter to our attention, and we are confident that the corrected analyses will rectify any misinterpretation.
- Explain how the results of the wound healing evaluation would likely predict the results for traditional methods of wound healing, such as in vivo wound healing models. Said differently, does a positive finding with this method correlate to animal models of wound healing, etc.?
More information and explanation with citation were added in lines 358-384 and in 4.2.6. Cell Migration Assay section.
The wound-healing assay is a straightforward and cost-effective technique that represents one of the earliest approaches for investigating directional cell migration in vitro. This method effectively replicates the cellular migration observed during the natural process of wound healing in vivo. The fundamental procedure entails inducing a simulated "wound" within a cell monolayer, documenting images at the initiation and at predetermined intervals throughout the migration process, and subsequently analyzing the images to quantify the rate of cell migration [39]. The key information obtained from the wound healing assay typically centers on the rate of gap closure, acting as a measure for the speed of collective cell movement. A cell-free gap within a cell monolayer can be generated through either direct manipulation or physical exclusion. To mimic the wound, the prevalent method involves inducing a gap by gently scratching a fully confluent monolayer using a pipette tip, needle, or another sharp instrument. Termed the scratch assay, this approach is favored due to its cost-effectiveness and simplicity. Typically, a pipette tip is employed to create a scratch one well at a time [51].
Confluent fibroblasts plated in a 6-well plates were scratched with a sterile 200 µL pipette tip and treated as described in Section 4.2.1. The scratched area was monitored with an inverted optical microscope at 40× magnification (Nikon; Minato, Tokyo, Japan). Fibroblast migration was calculated using dedicated Wound Healing tool for ImageJ® software and its rate was calculated according to the following formula:
- Add a section to the discussion section regarding the generalizability and also about the limitations.
Improved in lines 427-440
however, the application of oils in cellular studies faces several notable limitations. Firstly, the variable chemical composition of plant oils, influenced by factors such as source, cultivation conditions, and extraction methods, can introduce fluctuations in research outcomes. Ensuring oil purity, especially in naturally obtained oils, poses a challenge, with impurities potentially yielding unpredictable effects in cell experiments. Additionally, the susceptibility of some plant oils to oxidation requires careful attention to stability and proper storage to preserve their biological properties. Interactions between oils and other components in cell culture may influence experimental outcomes, necessitating consideration in data interpretation. Standardization becomes crucial due to variations in oil composition, aiding in the comparison of results across different experiments. Lastly, while plant oils often exhibit beneficial properties, their effects can be unpredictable depending on cell types and culture conditions. Awareness of these limitations is essential for researchers to adapt protocols effectively and ensure the reliability and reproducibility of their results.
- Discuss next steps so readers can understand what needs to be done to evaluate the effects in vivo and/or in vitro systems that model full thickness skin.
Improved in lines 358-384
Wound healing involves well defined repair stages, encompassing inflammation, proliferation, reepithelialization, and remodeling. Various preclinical models, including those involving mice, rabbits, and pigs, offer the ability to simulate acute or impaired wounds, such as those associated with diabetes or nutritional deficiencies. These models can be induced through diverse techniques, with excision or incision being the most prevalent. Once a suitable model is identified for a particular study, researchers must carefully choose methods that are both appropriate and reproducible, enabling the systematic monitoring of wound progression over time. Evaluation methods can range from non-invasive protocols like wound tracing, photographic documentation (with subsequent image analysis), and biophysical techniques, to more invasive protocols requiring wound biopsies. The selection of these methodologies is crucial in ensuring accurate and comprehensive insights into the dynamics of wound healing under specific conditions. [37]. Animal models and in vitro assays have evolved into essential tools for researchers across various scientific disciplines. This stepwise approach ensures a thorough understanding of the potential efficacy and safety of interventions before they progress to human trials, facilitating the development of more effective wound care strategies. In vitro studies are essential for assessing the potential efficacy of concentrations and evaluating the effectiveness of products on different cell types like fibroblasts and keratinocytes. They offer a cost-effective, rapid, and convenient approach for researchers. Notably, these assays provide valuable results swiftly and are ethically appealing as they typically do not involve the use of animals or humans. In the realm of wound healing research, in-vitro assays are particularly valuable for assessing the potential effectiveness of various treatments, including antimicrobial and wound-healing enhancers [38]. The wound-healing assay is a straightforward and cost-effective technique that represents one of the earliest approaches for investigating directional cell migration in vitro. This method effectively replicates the cellular migration observed during the natural process of wound healing in vivo [39].
- Because AmO is a complex mixture, how would one determine the ingredients (or combination of ingredients) responsible for the observed effects?
Improved in lines 441-459
Vegetable oils are distinguished by their chemical composition, which plays a pivotal role in determining their health properties. Analyzing characteristic chemical compositions, including fatty acid composition, tocols, phytosterols, squalene, carotenoids, phenolics, and phospholipids, provides insights into the diverse health benefits associated with various vegetable oils. These benefits encompass antioxidant activity, prevention of cardiovascular disease (CVD), anti-inflammatory, anti-obesity, anti-cancer properties, diabetes treatment, and kidney and liver protection. The exploration of health benefits is rooted in the predominant components found in representative oils. Each type of vegetable oil exhibits a unique chemical composition, with significant variations in key components, emphasizing the distinct advantages and health effects associated with different oils [42]. In this context, Amaranthus cruentus L. seed oil (AmO) stands out with a favorable chemical composition, featuring approximately 60% linoleic acid, 8-10% squalene, and tocopherols. Moving beyond health properties, the next crucial aspect of AmO's advantages lies in its chemical compositions, characterized by physicochemical properties such as melting point, flash-point, density, total acid number, iodine value, and saponification number. Qualitative and quantitative chromatographic analyses of phenolic compounds, determination of total antioxidant content (TAC), and identification of fatty acids and squalene further contribute to unraveling the comprehensive profile of AmO [43]. Therefore, AmO attracts attention as a unique mixture of many biologically active substances.
We thank the Reviewer for all these insightful suggestions.

Round 2
Reviewer 2 Report
Comments and Suggestions for Authors
The authors have greatly improved the manuscript with the changes. There are some remaining questions regarding the statistics but they should be easily addressed. To do so may require additional language in results and discussion. The attached comments explain the comparisons of interest and why they may be of value to the paper.

Author Response
January 04, 2024
Dr. Jakub Rok
Prof. Dr. Dorota Wrześniok
Special Issue Editors
International Journal of Molecular Sciences
Section: Molecular Pathology, Diagnostics, and Therapeutics,
Special issue: Radiation-Induced Damage to Human Skin: Biological and Medical Implications
Thank you for your letter of January 3, 2024, suggesting revision and resubmission of our manuscript (reference: Manuscript ID: ijms- 2787783) entitled, ”Amaranthus Cruentus L. Seed Oil Counteracts UVA - Radiation – Induced Inhibition of Collagen Biosynthesis and Wound Healing in Human Skin Fibroblasts” for publication in the International Journal of Molecular Sciences.
We found the comment and suggestion of the Reviewer helpful and have made revisions accordingly.
We hope that the revised manuscript has satisfied the Reviewer. We thank you for consideration and await your decision.
The response to the Reviewer and a new version of the manuscript with blue labelled revised sentences are attached.
Yours Sincerely,
Arkadiusz Surazynski, PhD.
Department of Medicinal Chemistry,
Medical University of Bialystok,
Kilinskiego 1,
Bialystok 15-089, Poland
e-mail:arkadiusz.surazynski@umb.edu.pl
Comments and Suggestions for Authors
The authors have greatly improved the manuscript with the changes. There are some remaining questions regarding the statistics but they should be easily addressed. To do so may require additional language in results and discussion. The attached comments explain the comparisons of interest and why they may be of value to the paper.
The Reviewer’s suggestion was addressed in the revised manuscript and the changes were marked in blue. Specifically in the results and discussion section, we improved the description of results in respect to the comparison between studied parameters versus control cells and UVA treated cells.
We again thank the Reviewer for all these insightful suggestions.
